# Pancreatic Cancer Cell-Derived Exosomes Promote Lymphangiogenesis by Downregulating ABHD11-AS1 Expression

**DOI:** 10.3390/cancers14194612

**Published:** 2022-09-23

**Authors:** Xulin Zhou, Fengyun Zhong, Yongmin Yan, Sihui Wu, Huizhi Wang, Junqiang Liu, Feifan Li, Dawei Cui, Min Xu

**Affiliations:** 1Department of Gastroenterology, Affiliated Hospital of Jiangsu University, Zhenjiang 212001, China; 2Department of General Surgery, The Second Affiliated Hospital of Soochow University, Suzhou 215025, China; 3Department of Laboratory Medicine, Wujin Hospital Affiliated with Jiangsu University, Jiangsu University, Changzhou 213000, China; 4The First Affiliated Hospital, Zhejiang University School of Medicine, Hangzhou 310030, China

**Keywords:** pancreatic cancer, exosomes, lymphangiogenesis, ABHD11-AS1

## Abstract

**Simple Summary:**

Lymphatic metastasis of pancreatic cancer is an important factor leading to poor prognosis of patients. In order to explore the relevant mechanism, we designed research and found that pancreatic cancer cell-derived exosomes promote lymphangiogenesis by downregulating the ABHD11-AS1 expression. This finding provides a new therapeutic strategy for inhibiting lymphatic metastasis metastasis in pancreatic cancer.

**Abstract:**

Research on pancreatic cancer microbiomes has attracted attention in recent years. The current view is that enriched microbial communities in pancreatic cancer tissues may affect pancreatic cancer metastasis, including lymph node (LN) metastasis. Similar to carriers of genetic information between cells, such as DNA, mRNA, protein, and non-coding RNA, exosomes are of great importance in early LN metastasis in tumors, including pancreatic cancer. Our previous study showed that the long non-coding RNA ABHD11-AS1 was highly expressed in tissues of patients with pancreatic cancer, and was correlated with patient survival time. However, the role of ABHD11-AS1 in pancreatic cancer LN metastasis has rarely been studied. Hence, in this paper we confirmed that exosomes derived from pancreatic cancer cells could promote lymphangiogenesis in vitro and in vivo, and that the mechanism was related to the downregulation of ABHD11-AS1 expression in lymphatic endothelial cells, and to the enhancement of their ability to proliferate, migrate, and form tubes. These findings preliminarily show a new mechanism by which pancreatic cancer cells regulate peripheral lymphangiogenesis, providing a new therapeutic strategy for inhibiting LN metastasis in pancreatic cancer.

## 1. Introduction

Pancreatic cancer is one of the most lethal solid tumors occurring worldwide. In China, pancreatic cancer ranks tenth in incidence and sixth in mortality among all malignancies [1]. Despite recent advances in diagnosing and treating pancreatic cancer, the prognosis of patients with pancreatic cancer remains poor, with a 5-year survival rate of <5% [2]. As pancreatic cancer is prone to early metastasis, patients often miss the best time to seek medical attention after they are diagnosed. Lymph node (LN) metastasis is one of the important means of pancreatic cancer metastasis, and the premature formation of metastases also makes some treatments intolerable for patients. Following treatment, the likelihood of recurrence is also greatly increased [3]. Therefore, inhibiting early LN metastasis in pancreatic cancer is essential in pancreatic cancer treatment.

Enriched lymphatic vessels around solid tumors are often a means for tumor cells to enter peripheral lymphatic vessels and invade the peripheral lymphatic system via LN metastasis, causing distant metastasis. The regulatory effect of tumor cells on the generation of the peripheral lymphoid system has been studied in many tumors, including breast cancer [4], lung cancer [5], prostate cancer [6], and colon cancer [7]. Tumor cells secrete various cytokines to regulate the proliferation, migration, and tube-forming abilities of their surrounding endothelial cells, enriching the vasculature around tumor cells. The regulatory role of pancreatic cancer cells on their peripheral lymphangiogenesis has also been explored for many years. Various molecules are involved in the regulation of lymphangiogenesis in pancreatic cancer. Heparanase can promote lymphangiogenesis in pancreatic neuroendocrine tumors [8], KAI1 can inhibit LN metastasis in pancreatic cancer cells [9], and proteinase-activated receptor–2 can inhibit tumor cell-mediated lymphangiogenesis; however, these molecules have no direct regulatory effect on lymphatic endothelial cells [10]. miR-206 inhibits lymphangiogenesis in pancreatic cancer [11], and circNFIB1 interacts with miR-486-5p to regulate miR-486-5p/PIK3R1/VEGF-C and inhibit lymphatic proliferation, tube hyperplasia, and metastasis in pancreatic cancer [12]. Relevant in vivo experiments have contributed to the progress of research on pancreatic cancers. Current studies have shown that the simultaneous targeting of TGF-β/EGFR/HER2 can inhibit the proliferation of lymphatic vessels in pancreatic cancer models [13]. Moreover, lymphatic hyperplasia in pancreatic cancer is vital for lymphatic metastasis in pancreatic cancer; related molecular mechanisms have always attracted the attention of researchers.

Exosomes are extracellular vesicles with a diameter of 30–150 nm, and are rich in diverse biologically active molecules such as proteins, nucleic acids, and lipids. After being endocytosed and taken up by recipient cells, these exosomes can regulate the biological functions of recipient cells [14]. In the formation of a tumor microenvironment, tumor cell-derived exosomes have always been considered to be a key factor in the regulation of intercellular communication. According to studies, tumor cell-derived exosomes have certain regulatory effects on tumor cell proliferation, metastasis, and stemness maintenance [15,16,17]. In pancreatic cancer, exosomes are important for the early diagnosis of tumor patients [18]. However, in-depth studies on exosomes associated with lymphatic vessel proliferation in pancreatic cancer are not yet available. The long non-coding RNA (lncRNA) ABHD11-AS1 is highly expressed in many cancers, including pancreatic cancer [19], papillary thyroid cancer [20], epithelial ovarian cancer [21], non-small cell lung cancer [22], gastric cancer [23], and colon cancer [24]. In patients with pancreatic cancer, high ABHD11-AS1 expression in the serum and tissues of patients is correlated with a poor prognosis [25]. However, studies on the association of high ABHD11-AS1 expression with lymphangiogenesis in pancreatic cancer are unavailable. Furthermore, whether pancreatic cancer cell-derived exosomes regulate LN metastasis in pancreatic cancer via ABHD11-AS1 is also unclear.

Thus, we hypothesize that microbes in the digestive tract colonize the pancreas via the bile duct, and secrete exosomes to promote the formation of pancreatic cancer. Simultaneously, pancreatic cancer cell-derived exosomes enhance the proliferation, migration, and tube formation of pancreatic cancer cells of lymphatic endothelial cells by regulating ABHD11-AS1 expression and causing lymphatic proliferation around the tumor, thereby promoting lymphatic metastasis in pancreatic cancer. In this study, we showed that pancreatic cancer cell-derived exosomes could promote the proliferation, migration, and tube formation of lymphatic endothelial cells in vitro and in vivo; furthermore, pancreatic cancer cell-derived exosomes could downregulate ABHD11-AS1 expression, indicating a regulatory role of ABHD11-AS1 in tube formation in lymphatic cells. We explored a novel mechanism leading to abnormal lymphatic proliferation around pancreatic cancer, which could be a means to a new strategy for treating pancreatic cancer lymphatic metastasis.

## 2. Methods

### 2.1. Clinical Samples and Study Approval

A total of 42 tumor tissues from patients with pancreatic cancer were obtained from the Affiliated Hospital of Jiangsu University. These patients were deemed eligible if they had pathologically confirmed PCa. All experiments were conducted with the approval of the Committees for Ethical Review of Research involving Human Subjects at Jiangsu University. Informed consent was obtained from all participants before their sample collection.

### 2.2. Cell Lines and Cell Culture

The human PCa cell lines Patu8988 and BxPC-3, and the normal human lymphatic endothelial cell (HULEC), were purchased from the American Type Culture Collection (Rockville, VA, USA). Patu8988 and HULEC cells were cultured in Dulbecco’s Modified Eagle Medium (DMEM; Gibco, New York, NY, USA) supplemented with 10% FBS. HULEC cells were cultured in RPMI 1640 medium (Gibco) supplemented with 10% FBS in a humidified incubator with 5% CO_2_ at 37 °C.

### 2.3. Exosome Isolation, Characterization, and Treatment

Exosomes were purified from PC cell-derived conditioned media through the polymer precipitation method. The PC cells were cultured in DMEM supplemented with 10% fetal bovine serum. The exosomes in bovine serum were depleted via ultracentrifugation at 160,000× *g* at 4 °C for 16 h before use. After the designated amount of time, the conditioned media were collected and centrifuged at 10,000× *g* at 4 °C for 30 min, and the supernatant was filtered through a 0.22-micrometer filter (Millipore, Burlington, MA, USA), followed by ultracentrifugation at 2000× *g* for 30 min at 4 °C. The exosome concentration solution was added to a supernatant in the ratio of 1:4. This mixture was preserved at 4 °C for 12 h, followed by centrifugation at 1500× *g* at 4 °C for 30 min, with collection of the precipitate, filtration through a 0.22-micrometer filter, and then ultracentrifugation at 2000× *g* for 30 min at 4 °C. The exosome pellet was washed with calcium and magnesium-free phosphate-buffered saline (PBS), followed by a second round of ultracentrifugation at 2000× *g* for 30 min at 4 °C and resuspension in PBS. The number of exosomes was determined by the BCA assay.

### 2.4. Transmission Electron Microscopy

After washing in PBS, the exosomes were fixed in 1.5 M sodium cacodylate buffer (pH 7.4) and absorbed onto Formvar/carbon support film copper-mesh grids, and negatively stained with 2% (*w*/*v*) uranyl acetate. The sample was observed under a transmission electron microscope (TEM). Digital images were acquired with an AMT digital camera system.

### 2.5. Wound Healing Assay

HLECs at 90–95% confluence were serum starved in a 6-well plate for 24 h. Then, the cells were carefully scratched with sterilized pipette tips. The movement of HLECs was recorded under the DM IRE2 microscope every 6 to 18 h. Both the migration path and direction of HLECs were imaged by microscopy and analyzed with ImageJ software.

### 2.6. Transwell Migration Assay

Transwell migration assays were performed with 24-well Transwell chamber plates (Corning, NY, USA). HLECs (8 × 10^4^ cells/150 μL/well) were seeded in the upper compartment of the transwell chamber. The lower compartment was supplemented with 0.6 mL of the cell culture medium. After 12 h of incubation, the HLECs that had migrated through the polycarbonate membrane were fixed, stained, and enumerated. The images were analyzed with ImageJ software.

### 2.7. Colony Formation Assay

HLECs were trypsinized into a 6-well plate at a concentration of 1000 cells/well, and cultured for 14 days under standard high-glucose conditions. The culture medium was replaced whenever required. The colonies were fixed and stained. Visible colonies were enumerated manually.

### 2.8. Cell Counting Kit 8

Cell viability was evaluated using the Cell Counting Kit-8 (CCK-8), according to the manufacturer’s guidelines. For the cell proliferation assay, HLECs were seeded onto a 96-well plate (5000 cells/plate) with 400 μg/mL exosomes for 24, 48, and 72 h. These cells were then incubated with the CCK8 kit at 37 °C for 2 h. Finally, the absorbance at 450 nm was measured with a microplate reader, in order to evaluate the viability of the indicated cells.

### 2.9. Reverse Transcription-Quantitative PCR (RT-qPCR)

Total RNA was extracted from the cells or tissues using Trizol and the enzyme RNA extraction kit (Takara, RR820A, Beijing, China). RNA (1 μg) was reverse-transcribed with a reverse transcription kit (Takara). Quantitative real-time PCR (qRT-PCR) was performed with gene-specific primers (Biotechnology, Shanghai, China) on the 7500ABI Biological System Machine. The comparison threshold was employed to calculate the absolute mRNA number. The relative gene expression normalized to β-actin was determined using the 2^−ΔΔCT^ method. The primer sequences used for ABHD11-AS1 were as follows: forward: 5′-ATGAAGCCATTGCCAAGAAG-3′; reverse: 5′-GCCTCTCTCTGCAGC TGATT-3′T.

### 2.10. Western Blotting

This experiment was performed as per the standard procedure. Briefly, approximately 60 μg of the total proteins were loaded and resolved in sodium dodecyl sulfate-polyacrylamide gel electrophoresis, and then transferred onto a polyvinylidene difluoride (PVDF) membrane. We blocked the PVDF membrane with 5% nonfat milk. Next, the membrane was co-incubated with the corresponding antibodies. Eventually, the membrane was subjected to electrochemiluminescence (ECL) and exposed under appropriate conditions.

### 2.11. Immunofluorescent Staining

The HLECs stimulated with 400 μg/mL exosomes were planted on glass coverslips in a 24-cell plate. These cells were treated with 0.4% paraformaldehyde for 30 min, and then with 0.5% Triton X-100 for 15 min, followed by washing thrice with PBS to remove the residual Triton X-100. Then, 0.5% goat serum was used to treat the cells for 1 h, followed by the application of the corresponding antibodies. The nucleus was then stained with DAPI. The pictures were recorded using confocal microscopy.

### 2.12. Immunohistochemical and Fluorescence In Situ Hybridization (FISH)

For immunohistochemical staining for D2-40 and LYVE-1, the tumor sections were deparaffinized and rehydrated, and the endogenous peroxidase was inactivated by 3% H_2_O_2_ for 30 min. Next, the slides were immersed in preheated antigen retrieval solution (0.01 M, pH 6.0, citrate buffer) for 30 min. After blocking with 5% BSA, the tumor slides were probed with a primary antibody against D2-40 (Abcam, Cambridge, UK) and LYVE-1 (Abcam) overnight at 4 °C, and then incubated with biotin-conjugated anti-rabbit IgG and streptavidin-biotin. Finally, the sections of tumor tissues were visualized with the DAB horseradish peroxidase color development kit (Boster, Wuhan, China) and counterstained with hematoxylin. Positive actions were defined as those shown brown signals in the cell cytoplasm. A stained index (value, 0–12) was determined by multiplying the score for staining intensity with the score for positive area. The intensity was scored as follows: 0, negative; 1, weak; 2, moderate; and 3, strong. The frequency of positive cells was defined as follows: 0, less than 5%; 1, 5% to 25%; 2, 26% to 50%; 3, 50% to 75%; and 4, greater than 75%. For example, a specimen containing 75% tumor cells with moderate intensity (3 × 2 = 6) and another 25% tumor cells with weak intensity (1 × 1 = 1) receive a final score of 6 + 1 = 7. Relevant images were acquired using a pathological section scanner (3DHISTECH, Budapest, Hungary). The FISH kit (GenePharma, Shanghai, China) was used to perform the FISH assay. ABHD11-AS1 was labeled with the FAM probe, and the nucleus was stained with DAPI. The images were recorded using fluorescence microscopy. The fluorescence intensity was recorded and analyzed via ImageJ. The value of fluorescence intensity is defined as the expression of ABHD11-AS1.

### 2.13. ABHD11-AS1 cDNA and siRNA Transfection

Full-length human ABHD11-AS1 cDNA was constructed into vectors (pCDH-CMV-ABHD11-AS1 cDNA-EF1a-GFP-T2A-Puro). Small-interfering RNA (siRNA) targeting ABHD11-AS1 and control-scrambled siRNA were designed (Genepharma, Suzhou, China). Pancreatic cancer cells were seeded up to 80% confluence in a 6-well plate in triplicate. Lipofectamine 2000 (Life Technologies, Carlsbad, CA, USA) reagent was used for the transfection of the ABHD11-AS1 expression vector or synthetic siRNA oligos into pancreatic cancer cells. The cell RNA samples were harvested at 48 h after transfection and subjected to Western blotting or quantitative RT-PCR analysis.

### 2.14. Statistical Analyses

Data were presented as the means ± SDs (standard deviations). Statistical significance between the two groups was analyzed via Student’s *t*-test using GraphPad Prism version 5.0 software. One-way analysis of variance (ANOVA) followed by the Dunnett test was applied for the analyses of more than two groups. A two-sided *p* < 0.05 was considered to indicate statistical significance.

## 3. Results

### 3.1. Pancreatic Cancer Exosome Isolation and Purification

We followed a polymer precipitation method to extract exosomes from the cell supernatants. We collected the supernatants of pancreatic cancer cell lines Patu8988 and Bxpc3, as well as human normal pancreatic ductal epithelial cells H6C7, and performed exosome extraction and purification on the corresponding supernatants. The Patu8988-derived exosomes were named Patu-ex; the Bxpc3-derived exosomes were named Bxpc3-ex; and the H6C7-derived exosomes were named H6C7-ex. The Western blot assay results showed that exosome-specific marker proteins cluster of differentiation (CD)9, CD63 and CD81 were present in Patu-ex, Bxpc3-ex, and H6C7-ex (Figure 1A). CD63 and CD81 were also present in these exosomes as per the flow cytometry analysis (Figure 1D). The typical bilayer structure and size of 30–100 nm were confirmed through transmission electron microscopy and nanoparticle tracking analysis, respectively (Figure 1B,C). The purified exosomes were labeled with the fluorescent membrane tracer DIO (green) and cultured with the human-derived lymphatic endothelial cells, in order to observe their distribution in lymphatic endothelial cells. After 48 h, cells phagocytosing exosomes were detected using confocal microscopy (Figure 1E). These results suggest that the extracted exosomes have typical characteristics of exosomes, and can be successfully phagocytosed by lymphatic endothelial cells.

### 3.2. Exosomes Derived from Pancreatic Cancer Cells Promote the Proliferation and Migration of Lymphatic Endothelial Cells

In order to investigate whether pancreatic cancer cell-derived exosomes have a certain regulatory effect on the biological function of lymphatic endothelial cells, we co-cultured Patu-ex, Bxpc3-ex, and H6C7-ex with the lymphatic endothelial cells for 48 h, and the same amount of a PBS buffer was used as a control. The subsequent scratch, migration, CCK8, and plate cloning experiments were performed to detect the migration and proliferation of the lymphatic endothelial cells after co-culturing with the exosomes. The results indicated that the migration ability of the lymphatic endothelial cells that were treated with the pancreatic cancer cell-derived exosomes (Patu-ex and Bxpc3-ex) was enhanced compared with that of the control group, and to cells treated with H6C7-ex (Figure 2A,Ci,Cii). Similarly, the proliferation ability of the lymphatic endothelial cells stimulated by Patu-ex and Bxpc3-ex was also significantly improved (Figure 2B,Di,Dii). Thus, we concluded that exosomes derived from pancreatic cancer cells could promote the proliferation and migration of lymphatic endothelial cells.

### 3.3. Pancreatic Cancer Cell-Derived Exosomes Promote the Tube Formation Surrounding Lymphatic Endothelial Cells

After confirming that Patu-ex and Bxpc3-ex can promote the proliferation and migration of human lymphatic endothelial cells, we further analyzed tube formation ability and tube formation-related proteins, in order to determine whether exosomes derived from pancreatic cancer cells can promote the formation of lymphatic vessels surrounding endothelial cells. The stimulation method and grouping were performed as described above, and tube formation was observed 4 h after initiating the tube formation experiment. The results showed that the length of the tube formed by the lymphatic endothelial cells stimulated by Patu-ex and Bxpc3-ex was longer than that of the control cells (Figure 3Ai,Aii). The intracellular lymphatic vessel marker protein LYVE-1 was detected with an immunofluorescence assay, and the results showed that LYVE-1 levels in the lymphatic endothelial cells were significantly increased compared with those in the control group after stimulation with the exosomes derived from pancreatic cancer cells (Figure 3B). Finally, we detected the levels of intracellular tube formation-related proteins after exosome stimulation with Western blotting, and the results indicated that the levels of the tube formation-related proteins TIE-2, angiopoietin-2 (Ang-2), LYVE-1, prospero homeobox protein 1, and vascular endothelial growth factor C increased (Figure 3C) the relevant statistical analysis is in the Appendix A. Hence, we concluded that exosomes derived from pancreatic cancer cells could promote the angiogenesis of lymphatic endothelial cells.

### 3.4. Pancreatic Cancer Cell-Derived Exosomes Promote Tube Formation by Downregulating ABHD11-AS1

Based on the above experiments, we confirmed that exosomes derived from pancreatic cancer cells could improve the proliferation, migration, and tube formation of lymphatic endothelial cells. Then, we studied their molecular mechanism of action. In patients with pancreatic cancer, ABHD11-AS1 expression is associated with patient prognosis; hence, we speculate that ABHD11-AS1 also plays a role in peripheral lymphangiogenesis stimulation by pancreatic cancer cells. We found that ABHD11-AS1 expression was downregulated in patients with positive LN metastasis, and increased in patients with negative LN metastasis (Figure 4B,C). We co-cultured Patu-ex, Bxpc3-ex, and H6C7-ex with lymphatic endothelial cells for 48 h and used the same amount of the PBS buffer as a control. We performed qRT-PCR to detect intracellular ABHD11-AS1 expression. The results showed that ABHD11-AS1 expression in the lymphatic endothelial cells decreased after the action of Patu-ex and Bxpc3-ex (Figure 4A). Therefore, we proposed a new hypothesis, that after the stimulation of lymphatic endothelial cells by pancreatic cancer cell-derived exosomes, the biological function of lymphatic endothelial cells is regulated through downregulating the lncRNA ABHD11-AS1. In order to test this hypothesis, we overexpressed ABHD11-AS1 (Flag-ABHD11-AS1) via lentiviral transfection in lymphatic endothelial cells, and also set a negative control (vector). Simultaneously, in another group of lymphatic endothelial cells, we used siRNA to interfere with ABHD11-AS1 expression (si-ABHD11-AS1), and also set a negative control (si-NC). The tube-forming ability was tested. The results of the CCK8 assay indicated that the proliferation of the lymphatic endothelial cells decreased after ABHD11-AS1 overexpression, whereas the proliferation of the lymphatic endothelial cells increased after interfering with ABHD11-AS1 expression (Figure 5A,B). The migration and tube formation experiments also showed similar results. The migration of the lymphatic endothelial cells and the ability to form tubes decreased after ABHD11-AS1 overexpression, whereas the migration of the lymphatic endothelial cells and the ability to form tubes increased after interfering with ABHD11-AS1 expression (Figure 5C–F). Subsequently, we tested this hypothesis. We stimulated lymphatic endothelial cells overexpressing ABHD11-AS1 with Patu-ex labeled with the membrane red dye DIL and verified the proliferation, migration, and growth of these cells. The results showed that after exosome stimulation, the proliferation, migration, and tube formation inhibition caused by ABHD11-AS1 overexpression were reversed (Figure 6A–C). Thus, we confirmed the hypothesis that pancreatic cancer cell-derived exosomes affect the proliferation, migration, and angiogenesis of lymphatic endothelial cells by downregulating ABHD11-AS1 expression in lymphatic endothelial cells.

### 3.5. Pancreatic Cancer Cell-Derived Exosomes Can Promote Pancreatic Cancer Cell Proliferation and Lymphatic Metastasis In Vivo

After confirming that exosomes derived from pancreatic cancer cells can promote the proliferation, migration, and tube formation of lymphatic endothelial cells in vitro, we also performed in vivo experiments to verify these results. In pancreatic cancer patients with positive and negative lymphatic metastases, we collected tumor tissues from them to perform immunohistochemical experiments. In the tumor tissues, lymphatic vessels were observed to be less enriched, whereas in the adjacent tissues, lymphatic vessels were enriched to a lesser extent. The degree of enrichment was high (Figure 7A). We injected intratumoral exosomes after tumor formation in a nude mouse model. The results suggested that exosomes could promote the growth of cancerous tumors. The tumor bodies of these nude mice were significant in size and weight. The same volume of PBS was injected into the control group (Figure 7B,C). After slicing the tumor, LYVE-1 immunohistochemistry showed that the enrichment of lymphatic vessels in the exosome-injected tumor was higher than that in the control group (Figure 7D). Thus, we showed that pancreatic cancer cell-derived exosomes can promote pancreatic cancer proliferation and lymphatic metastasis in vivo.

## 4. Discussion

In this study, we explored the role and mechanism of pancreatic cancer cell-derived exosomes in regulating lymphatic endothelial cells to form tubes and promote lymphangiogenesis. We found that exosomes derived from pancreatic cancer cells could promote the in vitro and in vivo proliferation of lymphatic endothelial cells and lymphangiogenesis. Pancreatic cancer cell-derived exosomes promote lymphangiogenesis by downregulating ABHD11-AS1 expression in lymphatic endothelial cells. In contrast, our results showed that ABHD11-AS1 overexpression in the lymphatic endothelial cells led to decreased cell proliferation, migration, and tube-forming ability, whereas interfering with ABHD11-AS1 expression reduced cell proliferation, migration, and tube-forming ability. The ABHD11-AS1-overexpressing lymphatic endothelial cells increased, and their proliferation, migration, and tube formation abilities increased after their co-culturing with the exosomes, consequently promoting lymphangiogenesis.

Pancreatic cancer is one of the deadliest diseases worldwide, with a median survival time of only 3–6 months. Despite recent advances in diagnosing and treating pancreatic cancer, the prognosis of patients with pancreatic cancer remains unsatisfactory [26]. Microorganisms have certain regulatory effects on the occurrence and development of pancreatic cancer. Microorganisms that are present in the pancreas are probably involved in the development of chronic pancreatitis [27]. Lymphatic metastasis is one of the important means of tumor metastasis, and is one of the crucial reasons for poor patient prognoses. In patients with pancreatic cancer, early lymphatic metastasis often leads to missed opportunities for optimal treatment by the time patients present symptoms. The abnormally enriched lymphatic vessels around the tumor can promote the early LN metastasis of pancreatic cancer; hence, research on lymphatic vessel hyperplasia in pancreatic cancer has high clinical application value. However, studies on the regulatory role of microorganisms in the lymphatic metastasis of pancreatic cancer are scarce. Exosomes are vesicles secreted by cells that are important for information exchange between cells. In recent years, intestinal flora-derived exosomes have been confirmed to be involved in the occurrence and progression of gastrointestinal diseases. Intestinal flora-derived exosomes can mediate steatosis in hepatocytes carrying HMGB1 [28]. In inflammatory bowel disease, exosomes can regulate local inflammation. Microorganisms in a tumor microenvironment play a key role in the construction and transformation of the microenvironment for tumor progression [29]. During the progression of colon cancer [30], liver cancer [31], and pancreatic cancer [32], the early detection of microorganisms can be used as a marker of patient prognosis. However, the specific mechanism remains unclear. Moreover, in a pancreatic cancer microenvironment, exosomes are important information carriers. Exosomes from various sources have regulatory effects on various cells in a tumor microenvironment, ultimately forming a microenvironment that is suitable for tumor growth. Exosomes derived from pancreatic cancer cells promote the activation of pancreatic stellate cells and affect their regulation in a tumor microenvironment [33]. Exosomes can also promote polarity changes in macrophages, which in turn regulate inflammatory responses in a tumor microenvironment [34]. There are few previous studies on pancreatic cancer cell-derived exosomes and lymphangiogenesis. By co-culturing exosomes with lymphatic endothelial cells, we observed that the exosomes derived from pancreatic cancer could promote the proliferation, migration, and tube formation of the lymphatic endothelial cells. Subsequently, we detected the tube-forming-related proteins including LYVE-1, Ang2, and TIE-2. The results also showed that the exosomes derived from pancreatic cancer cells could enhance tube-forming ability. Similar results have been reported in other tumors, where exosome-derived miR-320b promoted lymphatic endothelial cell angiogenesis by regulating the AKT pathway in esophageal squamous cell carcinoma [35]. VASH1 downregulation in cervical squamous cell carcinoma-derived exosomes carrying miR-221-3p resulted in enhanced cell tube-forming ability [36]. Hepatoma cell-derived exosomes carrying miR-296 regulated EAG1/VEGFA pathway, affecting the tube-forming ability of cells [37]. Therefore, exosomes derived from pancreatic cancer cells can promote lymphangiogenesis in vitro. This finding is also consistent with the proliferative effect of other tumor-derived exosomes on lymphatic vessels.

Subsequently, we performed some experiments to understand the mechanism. In recent years, lncRNAs have been implicated in various cellular life activities, such as cell proliferation, apoptosis, and differentiation [38]. Recent studies have shown that the lncRNA ABHD11-AS1 can promote the proliferation and metastasis of various cancer cells, such as papillary thyroid cancer [39], cervical cancer [40], breast cancer [41], and colon cancer [42]. ABHD11-AS1 expression promoted pancreatic cancer development [19]. In our previous study, we reported that ABHD11-AS1 was highly expressed in pancreatic cancer tissues, and was associated with poor patient prognoses [25]. The effect of ABHD11-AS1 on lymphatic endothelial cells is currently unknown. In this study, we hypothesized that pancreatic cancer-derived exosomes carrying lncRNA ABHD11-AS1 could promote the angiogenesis of lymphatic endothelial cells. However, the qRT-PCR results showed that ABHD11-AS1 expression was downregulated in lymphatic endothelial cells stimulated by the exosomes. In addition, after ABHD11-AS1 overexpression in these lymphatic endothelial cells, their angiogenesis ability decreased. In contrast, after interfering with ABHD11-AS1 expression, the angiogenesis ability of the lymphatic endothelial cells increased. We further stimulated the ABHD11-AS1-overexpressing lymphatic endothelial cells with exosomes, in order to restore the decreased vascular capacity. Therefore, we confirmed that the pancreatic cancer-derived exosomes regulated the angiogenesis ability of the lymphatic endothelial cells by downregulating ABHD11-AS1 expression. These results suggest that the lncRNA ABHD11-AS1 can inhibit the angiogenesis of lymphatic endothelial cells, and pancreatic cancer exosomes can promote cell angiogenesis by downregulating ABHD11-AS1 expression in lymphatic endothelial cells.

Finally, we confirmed the above results in vivo. In clinical tissues, by labeling lymphatic vessels, we could see that their enrichment degree in paracancerous tissues was higher than that in cancerous tissue. Areas with fewer lymphatic vessels were also observed. Subsequently, we established a nude tumor mouse model and injected the exosomes into the tumor. The results showed that the tumor became larger and heavier after exosome injection, and surrounding lymphatic vessels became more enriched. Exosomes can promote the proliferation of lymphatic vessels in pancreatic cancer. According to other studies, the generation of pancreatic cancer cell-derived exosomes induced under hypoxic conditions can promote microvascular angiogenesis in vivo [43]. Under normal conditions, pancreatic cancer cell-derived exosomes promote angiogenesis by regulating the CCAT1/miR-138-5p/HMGA1 axis [44]. However, under in vivo conditions, whether pancreatic cancer cell-derived exosomes can promote the proliferation of lymphatic vessels around pancreatic cancer, is still unknown.

This study also has limitations. Firstly, we did not fully explore the specific regulatory cellular pathways of exosomes derived from pancreatic cancer cells that promote lymphatic vessel proliferation. Furthermore, the specific downstream molecules of ABHD11-AS1 in lymphatic endothelial cells were not thoroughly studied. Secondly, early molecular typing to understand pancreatic cancer has recently attracted attention; despite this, we did not focus on whether molecular typing and pathological differences in cancer types would specifically affect the promotion of lymphatic vessel proliferation. Therefore, performing related studies is necessary.

## 5. Conclusions

We confirmed that pancreatic cancer cell-derived exosomes promote cell proliferation, migration, and angiogenesis by downregulating the expression of the lncRNA ABHD11-AS1 in lymphatic endothelial cells, thereby promoting lymphangiogenesis in pancreatic cancer cells. This finding provides a new therapeutic target for pancreatic cancer lymphatic metastasis, and a new strategy for the prognosis of patients with pancreatic cancer.

## Figures and Tables

**Figure 1 cancers-14-04612-f001:**
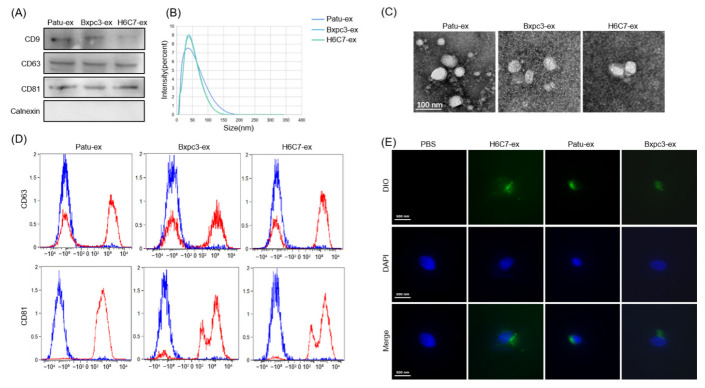
Identification of exosomes from pancreatic cancer cells. (**A**) The marker proteins of exosomes CD9, CD63, and CD81 were tested via Western blotting. (**B**) The size of exosomes was detected by the nanoparticle tracking analysis (NTA) assay. (**C**) The form of exosomes was observed under a transmission electron microscope (TEM). (**D**) Flow cytometry was performed to determine the marker proteins CD63 and CD81. Exosome samples were labeled in red and PBS was labeled in blue. (**E**) After dying with DIO, the exosomes were taken up by human lymphatic endothelial cells (HLEC). The pictures were taken with a confocal laser scanning microscope.

**Figure 2 cancers-14-04612-f002:**
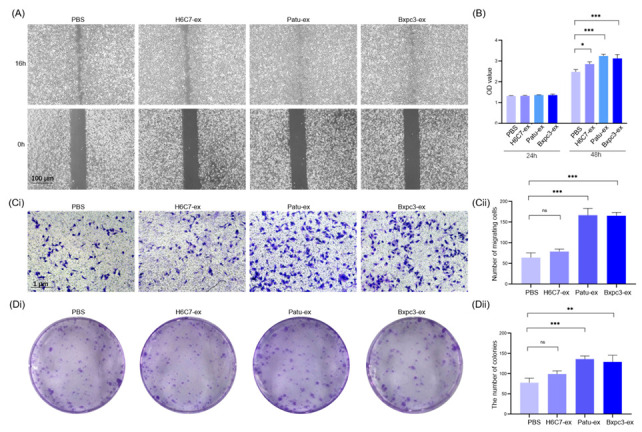
Exosomes from pancreatic cancer cells enhance the proliferation and migration of HLECs. (**A**) Wound healing assay was performed to watch the migration of HLECs in 16 h. (**B**) CCK8 assay was performed to test the viability of HLECs. (**Ci**) Transwell assay was performed to observe the number of transferred cells. Scale bar: 1 μm, 10×. (**Cii**) Statistical analysis of Transwell assay (*n* = 3). (**Di**) A colony formation assay was performed to observe the proliferation of HLECs. (**Dii**) Statistical analysis of the colony-formation assay (*n* = 3). * *p* < 0.05. ** *p* < 0.01, *** *p* < 0.001.

**Figure 3 cancers-14-04612-f003:**
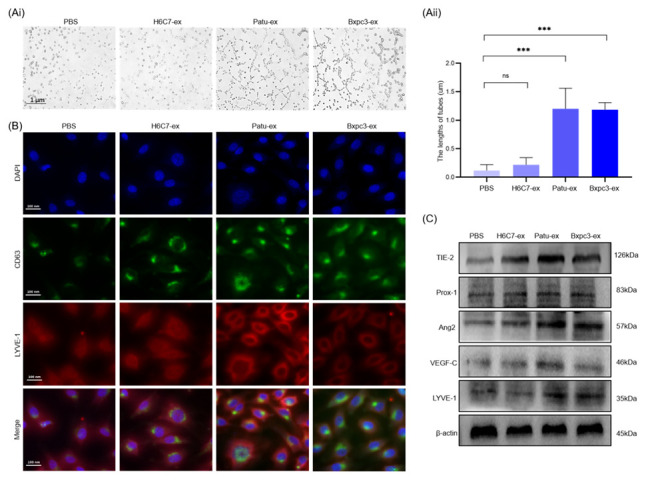
Exosomes from pancreatic cancer cells enhance lymphangiogenesis. (**Ai**) The tube-formation assay was conducted. (**Aii**) Statistical analysis of the tube-formation assay (*n* = 3). (**B**) Immunofluorescence assay was performed to test the specific protein of the lymphatic vessel, LYVE-1, and the marker protein of exosomes, CD63, in the HLECs after cultivating with the exosomes. (**C**) The relative tube-formation protein in the HLECs, TIE-2, Ang2, LYVE-1, Prox-1, and VEGF-C. *** *p* < 0.001.

**Figure 4 cancers-14-04612-f004:**
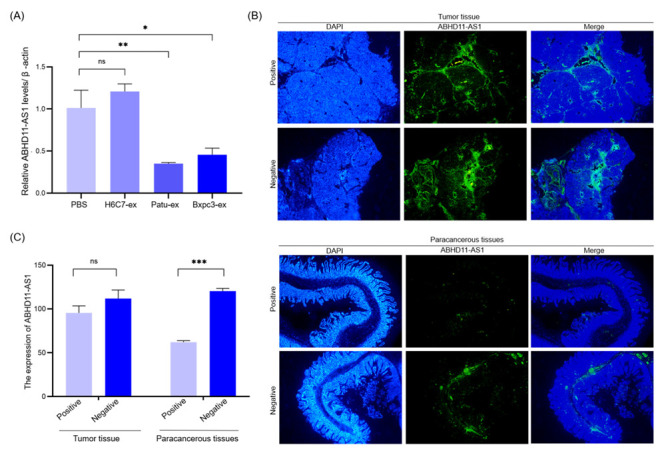
Exosomes from pancreatic cancer cells suppress the expression of ABHD11-AS1 in HLECs. (**A**) The relative expression of ABHD11-AS1 in HLECs after cultivating with exosomes (400 μg/mL) was tested with real-time PCR (*n* = 3). (**B**) The expression of ABHD11-AS1 was detected in tumor tissues and paracancerous tissues by the FISH assay. (**C**) Statistical analysis of the FISH assay (*n* = 3). * *p* < 0.05. ** *p* < 0.01, *** *p* < 0.001.

**Figure 5 cancers-14-04612-f005:**
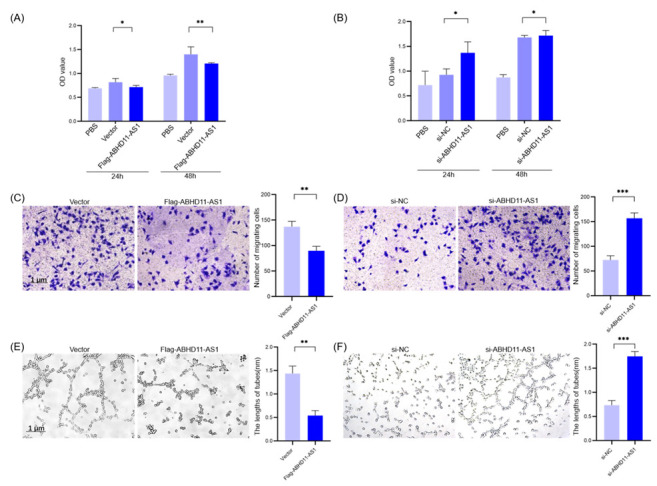
ABHD11-AS1 suppresses the proliferation, migration, and tube-forming ability of HLECs. ABHD11-AS1 was overexpressed or interfered with in the HLECs, and the proliferation and tube-formation of HLECs were tested. (**A**) CCK8 assay was performed to observe the viability of cells after overexpressing with ABHD11-AS1 (*n* = 3). (**B**) Viability of cells after interfering with ABHD11-AS1 was detected (*n* = 3). (**C**) Transwell assay was performed to detect the migration of HLECs after overexpressing with ABHD11-AS1 (*n* = 3). (**D**) After interfering with ABHD11-AS1, the migration of cells was tested (*n* = 3). (**E**) Tube-formation assay was performed to test the tube-forming ability of HLECs after overexpressing with ABHD11-AS1 (*n* = 3). (**F**) After interfering with ABHD11-AS1, the tube-forming ability of HLECs was tested (*n* = 3). * *p* < 0.05. ** *p* < 0.01, *** *p* < 0.001.

**Figure 6 cancers-14-04612-f006:**
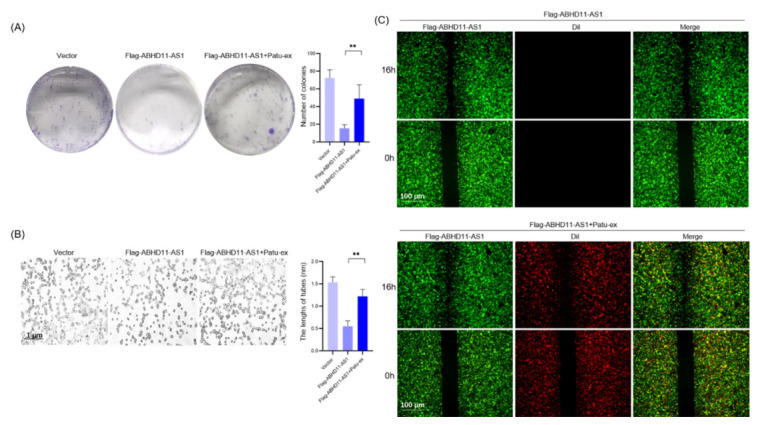
Exosomes from pancreatic cancer cells reversed the inhibitory effect of ABHD11-AS1 on HLECs. The stable HLECs with the overexpression of ABHD11-AS1 were fostered. These cells were cultivated with Patu-ex (400 μg/mL) for 48 h. (**A**) Colony formation assay was conducted to observe the proliferation of these cells (*n* = 3). (**B**) Tube-formation assay was conducted to test the tube-forming ability of these cells (*n* = 3). (**C**) Exosomes from Patu8988 were dyed with DIL, and the wound healing assay was conducted to observe the migration of these cells. ** *p* < 0.01.

**Figure 7 cancers-14-04612-f007:**
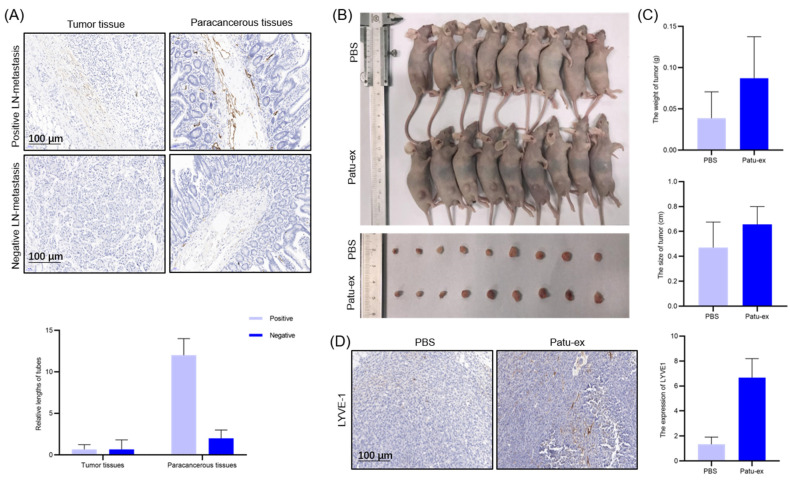
Exosomes from pancreatic cancer cells enhance the proliferation and lymphangiogenesis in vivo. (**A**) Immunohistochemical assay was conducted to examine the expression of D2-40, one of the specific proteins in the tumor tissue slices. Statistical analysis of immunohistochemical assay on the tube length in tumor and para-carcinoma tissues. (**B**) Pancreatic cancer cells were planted in a mouse model and categorized into 2 groups: injected with PBS (*n* = 9) or labeled with PBS and injected with PC-ex (*n* = 9) (20 mg/kg). Tumor tissue was collected after 2 weeks. (**C**) Statistical analysis of the tissue size and weight. (**D**) Slices of the tumors were subjected to an immunohistochemical assay to examine the expression of LYVE-1. Statistical analysis of immune-histochemical assay on the expression of LYVE-1 on the tissues obtained from the mouse model.

## Data Availability

All data generated or analyzed during this study are included in this published article.

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
