# Peer review of "Pancreatic Cancer Cell-Derived Exosomes Promote Lymphangiogenesis by Downregulating ABHD11-AS1 Expression"

_cancers, 2022, doi:10.3390/cancers14194612_

Round 1

Reviewer 1 Report

Review report for the manuscript entitled “Pancreatic cancer cell-derived exosomes promote lymph angiogenesis by downregulating ABHD11-AS1 expression

I have reviewed the manuscript on technical aspects. I personally feel that the authors should address the following comments and rewrite the entire results section for unbiased scientific review of the manuscript

11.The manuscript requires significant editing for better appreciation. (For example: line 88, line 250).

22.The authors are suggested to include the biological replicates (n) in the legends.

33. Check for HULEC and HLEC throughout the manuscript

44. The results section is hard to follow as the figure numbers are misrepresented throughout the manuscript.

55. Figure  1

1B- What are x and y axis,

1D, label the peaks for unstained, control, IgG control and CD63, CD81.

66. Figure 2Cii, X-axis should be number of migrating cells. Also there is no information about number of fields used for analysis, n and magnification.

77. Figure 3C: The authors should include quantification for the immunoblots TIE2, Prox-1, Ang2, VEGF-C, LYVE-1 in n=3.

88. Figure 4A, X-axis should be relative mRNA levels/b-actin

99. The authors should explain how the expression of ABHD11-AS1 was determined in Figure 4C. Represent the graph in fluorescence intensity or number of ABHD11-AS1 positive cells.

110. Figure 5C, 5D X-axis should be number of migrating cells

111.Figure 7D, The authors should explain how the expression of LYVE1 was determined in Figure 4C. Represent the graph as LYVE positive cells.

112. The role of microbiome is beyond the scope of this study as there are no data or experiment related to gut flora and pancreatic exosome. The authors should consider rewrite the introduction and discussion in the context of microbiome.

Author Response

Response to Reviewer 1 comments

1.The manuscript requires significant editing for better appreciation. (For example: line 88, line 250).

Response 1: We apologize for the language problems in the original manuscript. The language presentation was improved with assistance from a native English speaker with appropriate research background. We really hope that the flow and language level have been substantially improved. The revisions are marked in red in the uploaded revised manuscript

2.The authors are suggested to include the biological replicates (n) in the legends.

Response 2: We are grateful for the suggestion. In this study, our experiments were repeated three times, and the experimental results were statistically analyzed. We have added biological replicates (n=3) in the legends. The revisions are marked in red in the uploaded revised manuscript.

  1. Check for HULEC and HLEC throughout the manuscript

Response 3: We apologize for this problem in the original manuscript. We have uniformly abbreviated human lymphatic endothelial cells as HLECs, and the revisions are marked in red in the uploaded revised manuscript.

  1. The results section is hard to follow as the figure numbers are misrepresented throughout the manuscript.

Response 4: We apologize for misrepresenting the numbers of figures in the original manuscript. We have corrected the number of figures in revised manuscript. The result of Pancreatic cancer exosome isolation and purification is shown in figure 1. The result of Exosomes derived from pancreatic cancer cells promote the proliferation and migration of lymphatic endothelial cells is shown in figure 2 and figure 3. The result of Pancreatic cancer cell-derived exosomes promote tube formation of lymphatic endothelial cells is shown in figure 4. The result of Pancreatic cancer cell derived exosomes promote tube formation by down-regulating lncRNA ABHD11-AS1 is shown in figure 5, figure 6 respectively. The result of Pancreatic cancer cell-derived exosomes can promote pancreatic cancer proliferation and lymphatic metastasis in vivo is shown in figure 7. The revisions are marked in red in the uploaded revised manuscript.

  1. Figure  1

1B- What are x and y axis,

1D, label the peaks for unstained, control, IgG control and CD63, CD81.

Response 5: We apologize for lack the sign of x and y axis in Figure 1 B. Figure 1 B shows the size of exosomes is detected via Nanoparticle Tracking Analysis (NTA) assay, The X-axis represents the diameter of exosomes (nm), while the Y-axis represents the intensity(percent) of exosomes in the test sample. In the uploaded manuscript, the X-axis and Y-axis signs in Figure 1 B have been added. Figure 4D shows marker protein CD63, CD81 were tested via Flow Cytometry. In the Figure 4D, the exosome sample was marked in red and the control group is marked in blue. We apologize that the experimental and control groups were not clearly marked in the original manuscript. We have labeled the peaks in the legends of Figure 1D. 

  1. Figure 2Cii, X-axis should be number of migrating cells. Also, there is no information about number of fields used for analysis, n and magnification.

Response 6: We agree with this suggestion. As suggested by the reviewer, we have corrected the X-axis as ‘number of migrating cells’ in Figure 2Cii. Scale bars were 1 um, experiments were repeated three times, and magnification was 10X. We have added this information in Figure 2Ci legends, the revisions are marked in red in the uploaded revised manuscript.  

  1. Figure 3C: The authors should include quantification for the immunoblots TIE2, Prox-1, Ang2, VEGF-C, LYVE-1 in n=3.

Response 7: We are grateful for the suggestion. To be more clearly and in accordance with the reviewer concerns, we have added relative quantification pictures of the bands in the revised Supplementary Material. The revised supplemental material has been uploaded.

  1. Figure 4A, X-axis should be relative mRNA levels/b-actin

Response 8: Thanks for the reviewer’s great suggestion. In Figure 4A, we have corrected the X-axis as ‘relative ABHD11-AS1 levels/b-actin’ in Figure 4A.

  1. The authors should explain how the expression of ABHD11-AS1 was determined in Figure 4C. Represent the graph in fluorescence intensity or number of ABHD11-AS1 positive cells.

Response 9: We are grateful for the suggestion. In Figure 4B, we have examined the expression levels of ABHD11-AS1 in tissues using FISH experiments, where the fluorescence intensity was analyzed with Image The value of fluorescence intensity is recorded and defined as the expression of ABHD11-AS1. This part is added in page 7, line 187-193.

  1. Figure 5C, 5D X-axis should be number of migrating cells

Response 10: We feel great thanks for the suggestion. We have corrected the X-axis as ‘number of migrating cells’ in Figure 5C, 5D.

11.Figure 7D, the authors should explain how the expression of LYVE1 was determined in Figure 4C. Represent the graph as LYVE positive cells.

Response 11: We apologize for lack the method of Immunohistochemical assay in original manuscript. In Figure 7D, we examined the expression of LYVE-1 in tumor tissues via Immunohistochemical assay. Methods for detecting and calculating LYVE-1 expression have been added in the methodological section in the revised manuscript. This part is added in page 8, line 196-197.

  1. The role of microbiome is beyond the scope of this study as there are no data or experiment related to gut flora and pancreatic exosome. The authors should consider rewrite the introduction and discussion in the context of microbiome

Response 12. We deeply appreciate the reviewer’s comments. In fact, we also found that the presentation of the microbiome in the manuscript was too cumbersome which might interfere with appreciation of manuscript. We have rewritten the section on microbiomes in the introduction and discussion. The revisions are marked in red in the uploaded revised manuscript

Reviewer 2 Report

The manuscript “Pancreatic cancer cell-derived exosomes promote lymphangiogenesis by downregulating ABHD11-AS1 expression” by Zhou et al. is an original article investigating a possible role of exosomes in the lymphangiogenesis in PDAC.

Generally, pancreatic cancer is one of the most aggressive tumour with very dismal prognosis, due to different factors, including early metastases formation. One of the way of possible tumor cell escape are lymphatic vessels. Therefore the topic of this article is of high clinical interest. Exosomos act as “messanger” among tumour cells, and the authors demonstrated (in vitro and in vivo) that exosomes derived from PDAC cells can promote lymphangiogenesis and the mechanism is related to the down-regulation of long non-coding RNA ABHD11-AS1 expression in lymphatic endothelial cells with subsequent enhancement of their ability to proliferate and migrate. The authors discuss also the limits of the present study (page 13).

The article contains several wonderful figures which are always helpful.

I would like to give my compliments to the authors for this study. In conclusion, I would recommend the Editor to accept the manuscript for publication in this Journal, in the present form.

Author Response

Thank you for your nice comments on our article. The language presentation was improved with assistance from a native English speaker with appropriate research background. We really hope that the flow and language level have been substantially improved. The revised article has been uploaded.